# Development and external validation of a prognostic tool for COVID-19 critical disease

**Daniel S. Chow**[1]*, **Justin Glavis-Bloom**[1], **Jennifer E. Soun**[1], **Brent Weinberg**[2], **Theresa Berens Loveless**[3], **Xiaohui Xie**[4], **Simukayi Mutasa**[5], **Edwin Monuki**[6], **Jung In Park**[7], **Daniela Bota**[8], **Jie Wu**[9], **Leslie Thompson**[9], **Bernadette Boden-Albala**[10], **Saahir Khan**[11,12], **Alpesh N. Amin**[12], **Peter D. Chang**[1,4]

1 Department of Radiological Sciences, University of California, Irvine, California, United States of America, 2 Department of Radiological Sciences, Emory University, Atlanta, Georgia, United States of America, 3 Department of Biomedical Engineering, University of California, Irvine, California, United States of America, 4 Department of Computer Science, University of California, Irvine, California, United States of America, 5 Department of Radiological Sciences, Columbia University Medical Center, New York, New York, United States of America, 6 Department of Pathology and Laboratory Medicine, University of California, Irvine, California, United States of America, 7 Sue and Bill Gross School of Nursing, University of California, Irvine, California, United States of America, 8 UCI Center for Clinical Research, University of California, Irvine, California, United States of America, 9 School of Biological Sciences, University of California, Irvine, California, United States of America, 10 Department of Population Health and Disease Prevention and Department of Epidemiology, University of California, Irvine, California, United States of America, 11 Division of Infectious Diseases, University of California, Irvine, California, United States of America, 12 Department of Medicine, University of California, Irvine, California, United States of America

* Chowd3@hs.uci.edu

**Data Availability Statement:** Data are available from the UCI Institutional Data Access / Ethics Committee (contact via Joy Chu at joy.chu@uci.edu) for researchers who meet the criteria for

## Abstract

### Background

The rapid spread of coronavirus disease 2019 (COVID-19) revealed significant constraints in critical care capacity. In anticipation of subsequent waves, reliable prediction of disease severity is essential for critical care capacity management and may enable earlier targeted interventions to improve patient outcomes. The purpose of this study is to develop and externally validate a prognostic model/clinical tool for predicting COVID-19 critical disease at presentation to medical care.

### Methods

This is a retrospective study of a prognostic model for the prediction of COVID-19 critical disease where critical disease was defined as ICU admission, ventilation, and/or death. The derivation cohort was used to develop a multivariable logistic regression model. Covariates included patient comorbidities, presenting vital signs, and laboratory values. Model performance was assessed on the validation cohort by concordance statistics. The model was developed with consecutive patients with COVID-19 who presented to University of California Irvine Medical Center in Orange County, California. External validation was performed with a random sample of patients with COVID-19 at Emory Healthcare in Atlanta, Georgia.

access to confidential data due to ethical restrictions involving potentially identifying information.

**Funding:** This study was funded by an internal award at the University of California, Irvine through the COVID-19 Basic, Translational, and Clinical Research Funding Opportunity. None of the authors received salary support from this award. The funders had no role in study design, data collection and analysis, decision to publish, or preparation of the manuscript.

**Competing interests:** Alpesh Amin reported serving as PI or co-I of clinical trials sponsored by NIH/NIAID, NeuroRx Pharma, Pulmotect, Blade Therapeutics, Novartis, Takeda, Humanigen, Eli Lilly, PTC Therapeutics, OctaPharma, Fulcrum Therapeutics, Alexion. He has served as consultant and/or speaker for BMS, Pfizer, BI, Portola, Sunovion, Mylan, Salix, Alexion, AstraZeneca, Novartis, Nabriva, Paratek, Bayer, Tetraphase, Achogen LaJolla, Millenium, Ferring, PeraHealth, HeartRite, Aseptiscope, Sprightly. This does not alter our adherence to PLOS ONE policies on sharing data and materials.

## Results

Of a total 3208 patients tested in the derivation cohort, 9% (299/3028) were positive for COVID-19. Clinical data including past medical history and presenting laboratory values were available for 29% (87/299) of patients (median age, 48 years [range, 21–88 years]; 64% [36/55] male). The most common comorbidities included obesity (37%, 31/87), hypertension (37%, 32/87), and diabetes (24%, 24/87). Critical disease was present in 24% (21/87). After backward stepwise selection, the following factors were associated with greatest increased risk of critical disease: number of comorbidities, body mass index, respiratory rate, white blood cell count, % lymphocytes, serum creatinine, lactate dehydrogenase, high sensitivity troponin I, ferritin, procalcitonin, and C-reactive protein. Of a total of 40 patients in the validation cohort (median age, 60 years [range, 27–88 years]; 55% [22/40] male), critical disease was present in 65% (26/40). Model discrimination in the validation cohort was high (concordance statistic: 0.94, 95% confidence interval 0.87–1.01). A web-based tool was developed to enable clinicians to input patient data and view likelihood of critical disease.

## Conclusions and relevance

We present a model which accurately predicted COVID-19 critical disease risk using comorbidities and presenting vital signs and laboratory values, on derivation and validation cohorts from two different institutions. If further validated on additional cohorts of patients, this model/clinical tool may provide useful prognostication of critical care needs.

## Introduction

The exponential spread of coronavirus disease 2019 (COVID-19) has revealed constraints in critical care capacity around the globe [1, 2]. While there are early indications that social distancing measures have resulted in decreased transmission (i.e., "flattening the curve"), there is concern that subsequent pandemic waves may occur. Accurate and rapid patient prognostication is essential for critical care utilization management. Early identification of patients likely to develop critical disease may facilitate prompt intervention and improve outcomes.

Early reports suggest severe disease and poor outcomes are associated with older age, male sex, and comorbidities including hypertension, diabetes, and coronary artery disease [3–6]. Recent case series from the United States and France have additionally reported obesity is associated with hospitalization and worse COVID-19 disease [7–10]. Several attempts have been made to develop prognostic models for COVID-19 disease, largely based on early data from patient cohorts in China [11–18]. These models have used demographic features, including age, sex, and comorbidities, and a limited set of laboratory values including lymphocyte count, lactate dehydrogenase (LDH), C-reactive protein (CRP), which have been reported to be associated with more severe disease [19, 20]. These initial models are of variable quality, with a high likelihood of biases and limited numbers of variables, and performance evaluation is limited by suboptimal reporting and limited validation [21].

This study describes the development and external validation of a multivariate regression model and associated clinical tool to predict risk of COVID-19 critical disease, presented utilizing TRIPOD (transparent reporting of a multivariable prediction model for individual prognosis or diagnosis) reporting guidelines [22].

## Methods

### Study design and population

After approval of the institutional review board of the University of California, Irvine Medical Center, a prognostic model was developed with data from a single-center retrospective observational cohort study of sequential patients with COVID-19 disease diagnosed by nucleic acid detection from nasopharyngeal or throat swabs at the University of California, Irvine Medical Center (UCI Health) from March 1, 2020 to April 31, 2020 (derivation cohort). UCI Health is a 411-bed academic medical center located in Orange County, California which performed outpatient, emergency department, and inpatient COVID-19 testing throughout the study period.

The model was validated with a separate retrospective observational cohort of patients with COVID-19 disease at Emory Healthcare (validation cohort). Emory Healthcare is a multi-hospital 1500-bed academic system located in Atlanta, Georgia which performed outpatient, emergency department, and inpatient COVID-19 testing throughout the study period. Patients in the validation cohort were randomly selected from a radiology database of patients who underwent imaging with a clinical concern for COVID-19 disease from March 12, 2020 to April 7, 2020 and were diagnosed with COVID-19 by nucleic acid detection from nasopharyngeal swabs.

Data was obtained by manual chart review of the electronic health record. Clinical and laboratory values were obtained from the earliest documented result at the time of presentation. If a specific laboratory value was not initially available, the value occurring in time closest after presentation was used. If no value was obtained for a patient during the admission, it was marked as "missing". Data collection and validation were performed in accordance with the Institutional Review Board at each institution. Only de-identified data was transmitted between institutions. This study followed the Transparent Reporting of a Multivariable Prediction Model for Individual Prognosis or Diagnosis (TRIPOD) reporting guidelines [22].

### Outcome

The primary outcome was the likelihood of critical disease, defined as meeting the criteria of ICU admission, ventilation, and/or death. The initial index date for each patient was the date of COVID-19 diagnosis. All patients had follow-up of outcomes for a minimum of 10 days.

### Statistical analysis

**Developing the prediction model.** We searched the literature for predictors of COVID-19 disease severity and identified the following candidate predictors: demographic characteristics (age, sex), presenting vital signs (temperature, heart rate, respiratory rate, systolic blood pressure, diastolic blood pressure, body mass index), past medical history (hypertension, diabetes, cardiovascular disease, coronary artery disease, asthma, chronic kidney disease, metabolic syndrome [as defined by consensus criteria] [23], and total number of these comorbidities), and presenting laboratory values (white blood cell count, lymphocyte percentage, serum creatinine, aspartate aminotransferase [AST], lactate dehydrogenase [LDH], C-reactive protein [CRP], procalcitonin, ferritin, troponin I, d-dimer, triglycerides, and high density lipoprotein [HDL]).

With regards to the criteria for including variables into the model, using a recursive feature selection technique, univariate statistical testing was applied to the cohort to identify the variables with greatest differences in distribution when stratified based on outcomes. Starting with the variable calculated to have the largest differences, additional variables are added to the

model, one-by-one, in order of significance based on univariate testing, until the model performance plateaus. In this study, the optimal number of variables based on this technique was chosen to be 13. It should be noted that model performance did not degrade with additional variables (e.g. instead overall performance reached a plateau); thus the total number of variables used in this study represents the minimum amount needed to approximate the performance models using arbitrarily large number of covariates.

Two-sided *t*-tests were used for continuous variables and Pearson's chi-squared test ($\chi$2) was used for categorical variables to assess for differences in each candidate predictor based on critical disease status. Based on these results and prevalence of candidate predictor, the top thirteen covariates were chosen and used to create multivariable logistic regression model. For missing data, median imputation was performed based on underlying critical disease status. Each covariate was independently normalized to a scale of [0, 1] based on minimum and maximum values present in the dataset. Normalization of the data to similar scales facilitates numeric stability during the algorithm training process and ensures that all variables are initialized with relatively equal contribution to the prediction. Patients missing more than 50% of data (e.g. having 6 or less variables) were excluded from analysis.

The model was implemented using L2 regularization and optimized using the limited memory Broyden–Fletcher–Goldfarb–Shanno (BFGS) technique. Finally, a Wald chi-squared test was used to evaluate the contribution from each variable.

**Validating the prediction model.** The predictive accuracy of the model was determined retrospectively in the external validation cohort with discrimination and calibration. For any given patient, missing data was imputed using population-derived median values from the training cohort. Additionally, all model inputs were clipped to the minimum and maximum values present in the training cohort. Model discrimination (i.e., the degree to which the model differentiates between individuals with critical and non-critical outcomes) was calculated with the C statistic. All analyses were conducted using the Python scikit-learn library (0.22.2) [24] and IBM SPSS Statistics Subscription, version 1.0.0.1012 (IBM Corp., Armonk, N.Y., USA).

**Developing a clinical tool.** A web-based application was created in Python using a Flask server (1.1.1) to facilitate clinical implementation of the trained model.

## Results

For the derivation cohort, a total of 3,208 COVID-19 tests were conducted over the study period, of which 9.3% (299/3208) were positive. Clinical data including past medical history and presenting laboratory values were available for 29.1% (87/299) patients (median age, 48 years [range, 21–88 years]; 64.4% [56/87] male). Demographic detail is provided in Table 1. (Fig 1). Most common comorbidities included obesity (35.6%, 31/87), hypertension (36.8%, 32/87), and diabetes (24%, 24/87). Critical disease was present in 24.1% (21/87).

Of a total of 40 patients in the validation cohort (median age, 60 years [range, 27–88 years]; 55% [22/40] male), critical disease was present in 65% (26/40). Most common comorbidities included obesity (53%, 21/40), hypertension (60%, 24/40), and diabetes (40%, 16/40). Characteristics between the derivation and validation cohorts were notable for increased prevalence of comorbidities in the validation cohort.

After feature selection, the following factors associated with greatest increased risk of critical disease were used in model training: age, gender, total number of comorbidities (which included cardiovascular disease, coronary artery disease, chronic kidney disease, asthma/chronic obstructive pulmonary disease, diabetes mellitus, hypertension, and obesity), BMI,

**Table 1. Patient data.**

| Variable | Critical (n = 21) | Non-Critical (n = 66) |
|---|---|---|
| **Demographics** | | |
| Age (yr) | 55 | 46.5 |
| Male (count, %) | 15 (71.4) | 42 (62.1) |
| **Presenting Vital Signs** | | |
| Respiratory Rate | 22 | 18 |
| Body mass index (kg/m$^2$) | 33.2 | 27.5 |
| **Comorbidities** | | |
| Total Number | 2.2 | 1.0 |
| **Laboratory Values** | | |
| White blood cell count (1000/mcl) | 8.8 | 6.1 |
| % Lymphocytes | 15.0 | 22.2 |
| Serum Creatinine (mg/dL) | 1.6 | 0.9 |
| Lactate Dehydrogenase (U/L) | 513.8 | 248.4 |
| Troponin I (ng/L) | 43.7 | 8.7 |
| Ferritin (ng/mL) | 1066.8 | 372.7 |
| Procalcitonin (ng/mL) | 1.2 | 0.2 |
| C-reactive protein (mg/dL) | 13.1 | 8.1 |

All statistics represent the mean value unless otherwise stated. See Table 2 for F-statistic and details regarding modeling weights.

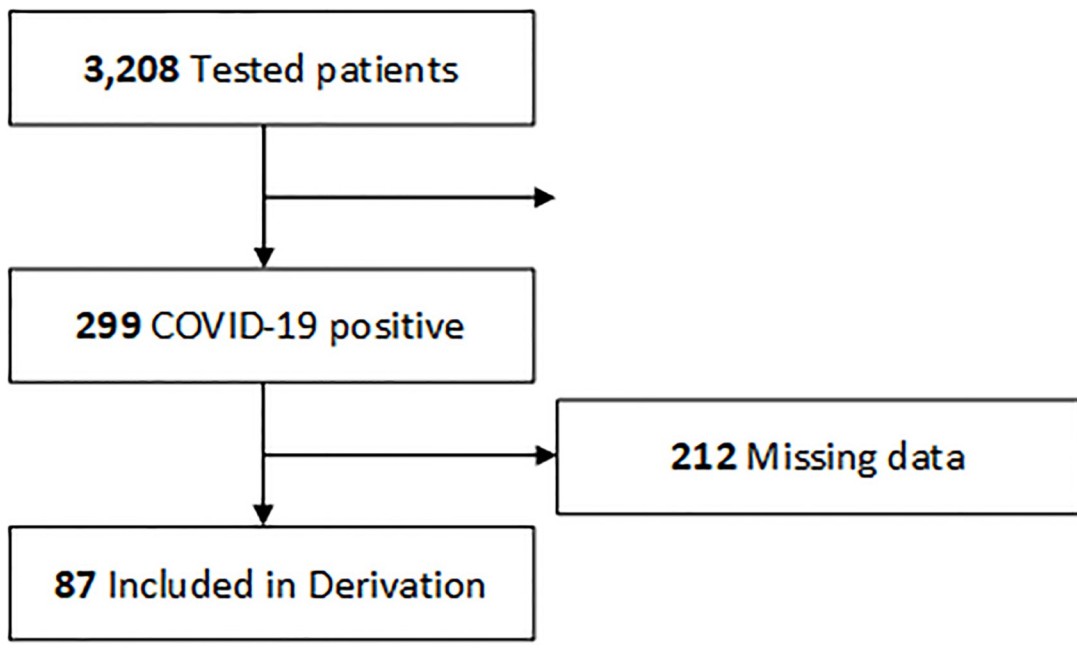

**Fig 1. Flow diagram of the derivation cohort.** For the derivation cohort, a total of 3,208 COVID-19 tests were conducted over the study period, of which 9.3% (299/3208) were positive. Of positive patients, laboratory data was available for 29.1% (87/299) patients.

**Table 2. Predictive model for COVID-19 critical disease.**

| Variable | Coefficient | Standard Error | Wald Statistic | f-test |
|---|---|---|---|---|
| **Demographics** | | | | |
| Age | 0.07 | 1.98 | 0.001 | 1.08 |
| Gender | 0.51 | 1.13 | 0.204 | 0.59 |
| **Presenting Vital Signs** | | | | |
| Respiratory Rate | 0.80 | 2.80 | 0.081 | 27.74 |
| Body mass index (BMI) | 1.07 | 2.49 | 0.185 | 13.91 |
| **Comorbidities** | | | | |
| Total Number | 1.14 | 1.62 | 0.491 | 16.80 |
| **Laboratory Values** | | | | |
| White blood cell count (WBC) | 0.14 | 2.68 | 0.003 | 7.87 |
| % Lymphocytes | -0.38 | 2.74 | 0.019 | 7.48 |
| Serum Creatinine | 0.24 | 4.75 | 0.002 | 6.89 |
| Lactate Dehydrogenase (LDH) | 1.72 | 2.12 | 0.658 | 23.13 |
| Troponin I | 0.60 | 3.38 | 0.032 | 7.26 |
| Ferritin | 0.55 | 3.17 | 0.030 | 12.38 |
| Procalcitonin | 0.67 | 4.38 | 0.023 | 7.59 |
| C-reactive protein (CRP) | 1.48 | 2.00 | 0.548 | 6.89 |

respiratory rate, white blood cell count, lymphocyte percentage, creatinine, lactate dehydroge-nase (LDH), troponin I, ferritin, procalcitonin, and C-reactive protein (CRP) (Table 2).

Model discrimination in the derivation cohort was high (concordance statistic: 0.948, 95% confidence interval 0.900–0.997);); with the best logistic regression score cut point at 30%, sensitivity was 90.4%, specificity was 89.4%, positive predictive value was 73.0%, and negative predictive value was 96.7%.

Model discrimination in the validation cohort was also high (concordance statistic: 0.940, 95% confidence interval 0.870–1.009); with the same 30% logistic regression cut point, sensitivity was 100%, specificity was 71.4%, positive predictive value was 86.7%, and negative predictive value was 100% (Fig 2). Procalcitonin was unavailable for all patients. During training, only 30/87 patients had all thirteen lab values present. The remaining patients had at least one missing variable, the distribution of missing variables is provided in Table 3. In our cohort, no patient requiring ICU admission had less than 6 lab values. The average number of missing variables was 1.33 (range 1–3) for cases that were correctly predicted as critical or non-critical and 2.11 (range 1–5) for cases that were incorrectly predicted.

A web-based tool was developed to enable clinicians to input patient data and view model output (Fig 3). The page accepts user input and outputs a likelihood of critical disease and does not require all variables to be present.

## Discussion

In this study, we developed and externally validated a predictive model and clinical tool that can be used to prognosticate the likelihood of COVID-19 critical disease based on data available early in a patient's presentation.

By using derivation and validation cohorts from separate institutions with different underlying patient characteristics, in particular a higher prevalence of comorbidities in the validation cohort, we achieved high calibration and discrimination. This model has the potential to be

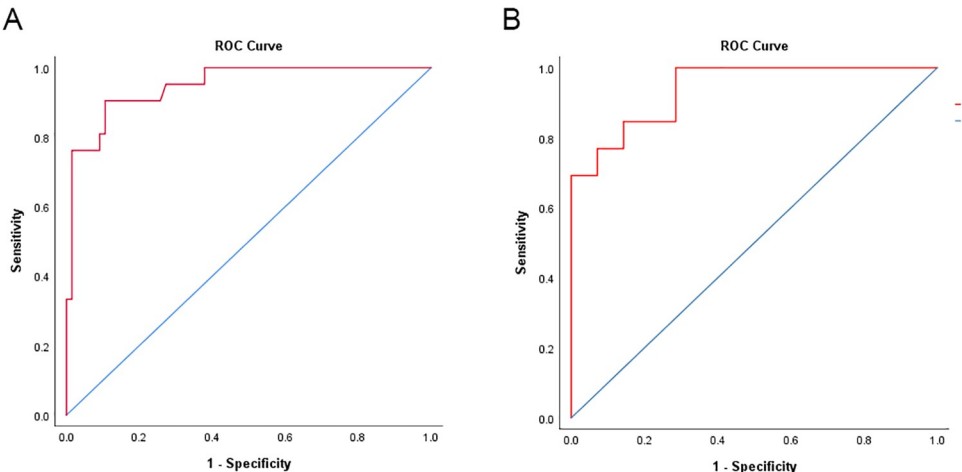

**Fig 2. Receiver operator curves.** Model discrimination for the derivation cohort (**A**) was (concordance statistic: 0.948, 95% confidence interval 0.900–0.997) and validation cohort (**B**) was (concordance statistic: 0.940, 95% confidence interval 0.870–1.009).

utilized by front-line healthcare providers to predict critical care demand and provide early indications of likelihood a patient's condition may worsen. As therapeutic interventions become validated, this may enable early intervention in at-risk patients to improve outcomes. In particular, antiviral therapies may have increased efficacy if administered earlier in the disease course.

Compared with other earlier models, which were primarily single institution-based, were developed from patient cohorts in China, utilized only a few variables, and did not include subsequently identified risk factors such as number of comorbidities and obesity [7–10], this model may have greater relevance and predictive strength in cohorts of Western patients in which obesity is more common. In particular, the inclusion of nearly 30 candidate variables in model derivation ensures sufficient consideration to numerous previously identified prognostic correlates.

Interestingly, variables which have previously been reported to be associated with worse COVID-19 disease, most notably including older age and hypertension, were less predictive in our sample than body mass index, total number of comorbidities and several laboratory values. The tool performed well in the validation set even though there was a higher rate of missing data for some values such as procalcitonin and ferritin, which were not frequently performed

**Table 3. Variable distribution for patients.**

| Number of Variables | Patient Count |
| --- | --- |
| 7 | 2 |
| 8 | 37 |
| 9 | 2 |
| 10 | 5 |
| 11 | 5 |
| 12 | 6 |
| 13 | 30 |

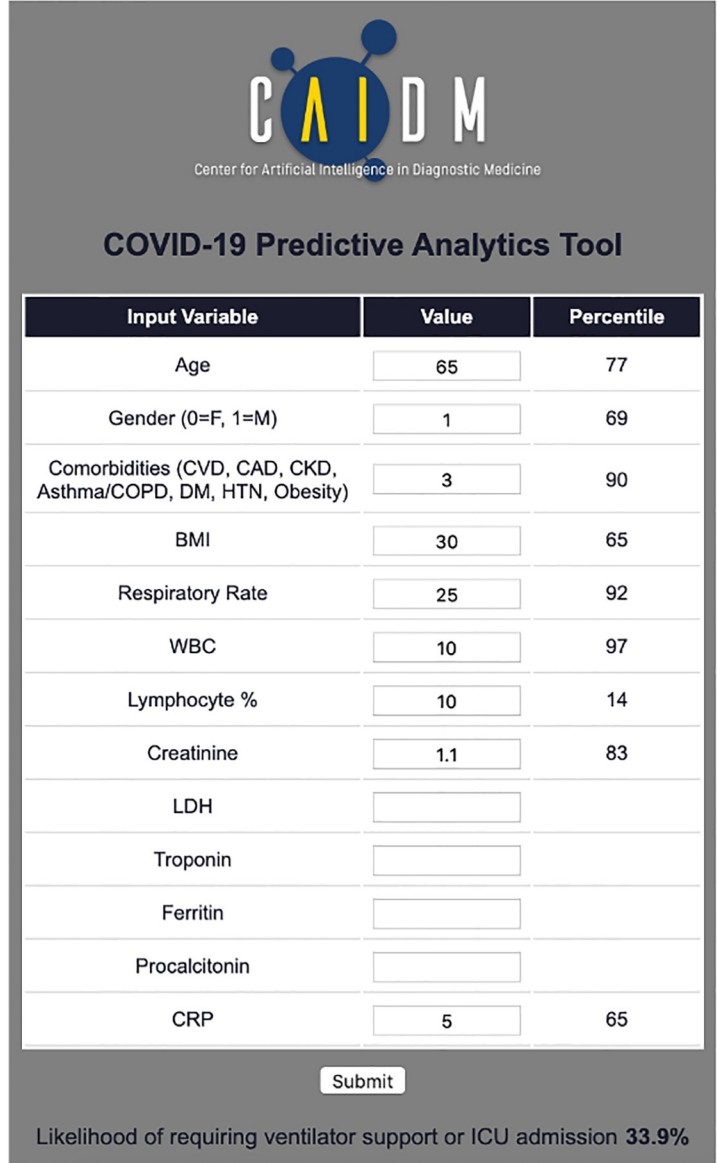

**Fig 3. Web-based clinical tool for COVID-19 critical disease prediction.**

at the validation institution. In settings in which laboratory data is easily and rapidly acquired, this study suggests there may be value to establishing a panel of COVID-19-specific laboratory studies including lactate dehydrogenase, troponin I, ferritin, procalcitonin, and C-reactive protein (in addition to commonly acquired complete blood count and serum chemistries).

Front-line medical providers have been inundated with critically ill COVID-19 patients. A simple web-based tool utilized at patient presentation may facilitate decision making by simplifying integration of numerous clinical variables. Our model has a high negative predictive value, which can increase physician confidence in determining which patients may be discharged safely at presentation. This is of particular utility in settings of high healthcare utilization, especially when physicians are treating higher than expected numbers of patients and/or

working outside of their standard practice. Our model has high positive predictive value, highlighting those patients for whom admission and close clinical monitoring may be appropriate.

The chosen cutoff point of 30% based on the derivation cohort performs with 100% sensitivity and 71.4% specificity in the validation cohort that included more critical patients. In most circumstances, identifying all cases of critical disease is preferred even if some less critical patients are identified, but in certain situation such as a surge in which critical care resources are limited, the cutoff point could be adjusted to a desired balance of sensitivity and specificity.

## Limitations

This study has limitations. A limited small sample of patient data was reviewed retrospectively from two centers. As data was obtained retrospectively, there was no control over which laboratory data was collected, which varied with institutional practice patterns. However, the model performed well in a validation data set with incomplete laboratory values. Further testing on larger cohorts of patient data is needed. Conclusions may not be globally generalizable to different patient cohorts. Lastly, not all patients had complete data available. While imputation is an imperfect approximation to true lab value data, the high performance on an external data set with missing data suggests that the approach is reasonable.

## Conclusions

We present a predictive model and clinical tool which can be used to prognosticate the likelihood of COVID-19 critical disease based on data at patient presentation. Further testing is needed on larger patient cohorts to establish generalizability. In subsequent analyses, we intend to evaluate whether this model can be applied to daily trends of clinical data in admitted patients to predict patient disposition.

## Author Contributions

**Conceptualization:** Daniel S. Chow, Justin Glavis-Bloom, Jennifer E. Soun, Theresa Berens Loveless, Xiaohui Xie, Edwin Monuki, Jung In Park, Leslie Thompson, Bernadette Boden-Albala, Saahir Khan, Alpesh N. Amin, Peter D. Chang.

**Data curation:** Daniel S. Chow, Justin Glavis-Bloom, Jennifer E. Soun, Brent Weinberg, Theresa Berens Loveless, Xiaohui Xie, Simukayi Mutasa, Edwin Monuki, Jung In Park, Daniela Bota, Jie Wu, Leslie Thompson, Saahir Khan, Alpesh N. Amin, Peter D. Chang.

**Formal analysis:** Daniel S. Chow, Justin Glavis-Bloom, Jennifer E. Soun, Brent Weinberg, Theresa Berens Loveless, Xiaohui Xie, Simukayi Mutasa, Edwin Monuki, Jung In Park, Daniela Bota, Jie Wu, Leslie Thompson, Bernadette Boden-Albala, Saahir Khan, Alpesh N. Amin, Peter D. Chang.

**Funding acquisition:** Daniel S. Chow, Justin Glavis-Bloom, Jennifer E. Soun, Xiaohui Xie, Edwin Monuki, Jung In Park, Daniela Bota, Leslie Thompson, Bernadette Boden-Albala, Alpesh N. Amin, Peter D. Chang.

**Investigation:** Daniel S. Chow, Justin Glavis-Bloom, Jennifer E. Soun, Brent Weinberg, Theresa Berens Loveless, Xiaohui Xie, Simukayi Mutasa, Edwin Monuki, Jung In Park, Daniela Bota, Leslie Thompson, Bernadette Boden-Albala, Saahir Khan, Alpesh N. Amin, Peter D. Chang.

**Methodology:** Daniel S. Chow, Justin Glavis-Bloom, Jennifer E. Soun, Brent Weinberg, Xiaohui Xie, Simukayi Mutasa, Edwin Monuki, Jung In Park, Daniela Bota, Jie Wu, Leslie Thompson, Bernadette Boden-Albala, Saahir Khan, Alpesh N. Amin, Peter D. Chang.

**Project administration:** Daniel S. Chow, Justin Glavis-Bloom, Jennifer E. Soun, Theresa Berens Loveless, Xiaohui Xie, Simukayi Mutasa, Edwin Monuki, Jung In Park, Daniela Bota, Leslie Thompson, Saahir Khan, Alpesh N. Amin, Peter D. Chang.

**Resources:** Daniel S. Chow, Justin Glavis-Bloom, Jennifer E. Soun, Brent Weinberg, Xiaohui Xie, Simukayi Mutasa, Edwin Monuki, Jung In Park, Daniela Bota, Jie Wu, Leslie Thompson, Bernadette Boden-Albala, Saahir Khan, Alpesh N. Amin, Peter D. Chang.

**Software:** Daniel S. Chow, Justin Glavis-Bloom, Jennifer E. Soun, Xiaohui Xie, Simukayi Mutasa, Edwin Monuki, Jung In Park, Jie Wu, Saahir Khan, Alpesh N. Amin, Peter D. Chang.

**Supervision:** Daniel S. Chow, Justin Glavis-Bloom, Jennifer E. Soun, Theresa Berens Loveless, Xiaohui Xie, Simukayi Mutasa, Edwin Monuki, Daniela Bota, Leslie Thompson, Bernadette Boden-Albala, Saahir Khan, Alpesh N. Amin, Peter D. Chang.

**Validation:** Daniel S. Chow, Justin Glavis-Bloom, Jennifer E. Soun, Brent Weinberg, Xiaohui Xie, Simukayi Mutasa, Edwin Monuki, Jung In Park, Daniela Bota, Jie Wu, Saahir Khan, Alpesh N. Amin, Peter D. Chang.

**Visualization:** Daniel S. Chow, Justin Glavis-Bloom, Jennifer E. Soun, Xiaohui Xie, Simukayi Mutasa, Edwin Monuki, Daniela Bota, Jie Wu, Saahir Khan, Alpesh N. Amin, Peter D. Chang.

**Writing – original draft:** Daniel S. Chow, Justin Glavis-Bloom, Jennifer E. Soun, Brent Weinberg, Simukayi Mutasa, Daniela Bota, Leslie Thompson, Bernadette Boden-Albala, Saahir Khan, Peter D. Chang.

**Writing – review & editing:** Daniel S. Chow, Justin Glavis-Bloom, Jennifer E. Soun, Brent Weinberg, Theresa Berens Loveless, Xiaohui Xie, Simukayi Mutasa, Edwin Monuki, Jung In Park, Daniela Bota, Leslie Thompson, Bernadette Boden-Albala, Saahir Khan, Alpesh N. Amin, Peter D. Chang.

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
