## [Decision Letter · Decision Letter 0]

30 Jul 2020

PONE-D-20-14464

Development and External Validation of a Prognostic Tool for COVID-19 Critical Disease

PLOS ONE

Dear Dr. Chow,

Thank you for submitting your manuscript to PLOS ONE. After careful consideration, we feel that it has merit but does not fully meet PLOS ONE’s publication criteria as it currently stands. Therefore, we invite you to submit a revised version of the manuscript that addresses the points raised during the review process.

Two reviewers commented on your study.  Their comments were not supportive in favor of accepting your study for publication.  Their specific comments are attached below and should be addressed if a revision is submitted.  I want to add one more comment that should be addressed.  This study was performed 4 months ago.  If your tool was used prospectively since then, this should be commented on, providing the observed reliability of the tool in identifying patients who will develop critical disease.  If this request exceeds what was authorized by the institution's research ethics committee, please use this letter when approaching the IRB to authorize a change in the protocol.  

We look forward to receiving your revised manuscript.

Kind regards,

Itamar Ashkenazi

Academic Editor

PLOS ONE

Journal Requirements:

2.Thank you for stating the following financial disclosure:

 [The funders had no role in study design, data collection and analysis, decision to publish, or preparation of the manuscript.].

3.We note that you have indicated that data from this study are available upon request. PLOS only allows data to be available upon request if there are legal or ethical restrictions on sharing data publicly. For information on unacceptable data access restrictions, please see http://journals.plos.org/plosone/s/data-availability#loc-unacceptable-data-access-restrictions.

Reviewers' comments:

Reviewer's Responses to Questions

**Comments to the Author**

1. Is the manuscript technically sound, and do the data support the conclusions?

Reviewer #1: Partly

Reviewer #2: Partly

2. Has the statistical analysis been performed appropriately and rigorously? 

Reviewer #1: Yes

Reviewer #2: I Don't Know

3. Have the authors made all data underlying the findings in their manuscript fully available?

Reviewer #1: Yes

Reviewer #2: No

4. Is the manuscript presented in an intelligible fashion and written in standard English?

Reviewer #1: Yes

Reviewer #2: Yes

5. Review Comments to the Author

Reviewer #1: The concept of developing a risk score for which Covid 19 patients will require critical care is laudable. Prediction of which patients will come to need such care is very moprtant for planning and allocation of resources.

The paper is written in clear and standard English.

The methodology is sound, however this reviewer would be grateful fi the authors could address the following:

The very limited sample size - 87 in the drivation cohort of which 21 wre critical and 40 in the validation cohort of which 26 were critical.

Why the derivation and validation cohorts were drawn from different insitutions, particulalry where the laboratory sampling in the validation group did not include important elements of the derivation group data.

What effect the retrospective data collection might have had on study results (this is briefly alluded to in the "limitations" section).

Reviewer #2: The authors made a prediction model for critical disease (defined as ICU admission, MV or death) in 87 COVID-19 patients and found a high C-statistic of 0.95 and 0.94 in the validation cohort and concluded that the model performed well.

I have some questions.

Effective sample size is 21 (out of 87), is it allowed to put 13 variables in the model (the full model consisted even more variables) while one variable (total number of co-morbidities consists another 8 variables)?

If it is allowed, for clinical practice it is not useful. A practical clinical prediction model consists of 3-4 variables that are ready available at the bedside. Of course does accuracy increase with more variables, but consider reducing the amount of variables.

Out of 299 patients, 212 were excluded due to missing variables. The fact that data are missing in these patient may not be random. Furthermore, even in de 87 eligible patients data was not complete (mean missing’s was 1.33(range 1-3) for every patient. Therefore imputation was performed. What were the results when missing’s considered as missing’s?

What is the necessity and effect of normalizing the data?

What were the criteria for including a variable in the model?

I miss a Table 1 with all baseline characteristics (and difference between yes/no critical disease) and perhaps a table 2 with univariate logistic regression analyses of these parameters.

6. PLOS authors have the option to publish the peer review history of their article (what does this mean?). If published, this will include your full peer review and any attached files.

Reviewer #1: **Yes: **PV van Heerden

Reviewer #2: **Yes: **Walter m. van den Bergh

---

## [Author Response · Author response to Decision Letter 0]

18 Sep 2020

Ref: PONE-D-20-14464

Dr. Ashkenazi and PLOS ONE Reviewers, 

Thank you for your review of our manuscript. We sincerely appreciate the comments provided. We have made edits to our manuscript based on these comments, addressing all of the issues that have been raised. In addition, we have responded directly to each of the reviewer remarks below. Please do not hesitate to contact us with additional comments or suggestions.

Reviewer #1: 

1. The concept of developing a risk score for which Covid 19 patients will require critical care is laudable. Prediction of which patients will come to need such care is very moprtant for planning and allocation of resources. The paper is written in clear and standard English.

Thank you for reviewing our manuscript and for your comments and suggestions.

2. The methodology is sound, however this reviewer would be grateful fi the authors could address the following:

a. The very limited sample size - 87 in the drivation cohort of which 21 wre critical and 40 in the validation cohort of which 26 were critical. Why the derivation and validation cohorts were drawn from different insitutions, particulalry where the laboratory sampling in the validation group did not include important elements of the derivation group data.

Early in the COVID-19 pandemic there were a limited number of patients from which to derive and validate our predictive model. Utilizing a cohort of patients from an outside institution was necessary, as we lacked sufficient numbers of patients at our institution to perform both derivation and validation. As laboratory variables were collected retrospectively, there were differences in clinical practice patterns which affected data availability. Additionally, the use of a completely external validation group increases the confidence in generalizability of the results. Though the external validation group lab values are not as comprehensive as the training set, the relatively robust performance suggests that the algorithm will nonetheless maintain accuracy even with missing data.

b. What effect the retrospective data collection might have had on study results (this is briefly alluded to in the "limitations" section).

The principal limitation is data availability. To overcome this limitation, we suggested the development of a panel of COVID-19 laboratory tests, which can help standardize clinical practice, and which we have subsequently implemented at our institution. 

Reviewer #2:

3. The authors made a prediction model for critical disease (defined as ICU admission, MV or death) in 87 COVID-19 patients and found a high C-statistic of 0.95 and 0.94 in the validation cohort and concluded that the model performed well.

Thank you for reviewing our manuscript and for your comments and suggestions.

4. Effective sample size is 21 (out of 87), is it allowed to put 13 variables in the model (the full model consisted even more variables) while one variable (total number of co-morbidities consists another 8 variables)?

Thank you for this suggestion. We agree that the use of many model parameters may increase the risk of model overfitting. Using a recursive feature selection process, one feature was added to the model at a time and the performance of the new model was assessed via a cross-validation technique. Despite the use of thirteen features, no significant overfitting was observed across each of the cross-validation folds during training. Furthermore, the high performance of the model on the external test set helps to validate this approach and give confidence to the use of all thirteen variables in the final predictive algorithm. 

Regarding the composite variable for comorbidity: a composite variable that captures the total number of 8 comorbidities demonstrated strong predictive value (Wald score 0.491). We hypothesize that this is because COVID-19 is a multiorgan/multisystem disease and that overall patient health status, as captured by the total number of comorbidities, is more important than a specific comorbid condition.

5. If it is allowed, for clinical practice it is not useful. A practical clinical prediction model consists of 3-4 variables that are ready available at the bedside. Of course does accuracy increase with more variables, but consider reducing the amount of variables. 

Thank you for this observation. While the full model can use up to 13 different variables, in fact there is no requirement that all variables must be present for either training of prediction. During training, only 30/87 patients had all thirteen lab values present. The remaining patients had at least one missing variable, the distribution of which is shown here:

Number of variables Total patient count

7 2

8 37

9 2

10 5

11 5

12 6

13 30

During the prediction process, all missing data is accounted for using median imputation from population statistics from the training data. While imputation is an imperfect approximation to true lab value data, the high performance on an external data set with missing data suggests that the approach is reasonable. Additionally, as a surrogate for clinical utility, the clinical prediction model has been integrated into the clinical workflow at our hospital and is furthermore available as a public website at http://covidrisk.hs.uci.edu (Figure 3). 

This table and added has been added to the results of the manuscript as well as the discussion in the limitation section.

Out of 299 patients, 212 were excluded due to missing variables. The fact that data are missing in these patient may not be random. Furthermore, even in de 87 eligible patients data was not complete (mean missing’s was 1.33(range 1-3) for every patient. Therefore imputation was performed. What were the results when missing’s considered as missing’s?

The distribution of data availability per patient is shown in the table above. Patients missing more than 50% of data (e.g. having 6 or less variables) were excluded from analysis. In our cohort, no patient requiring ICU admission had less than 6 lab values.

This discussion has been added to the methods and results where appropriate.

We agree that these patients may not be random. In general, such patients have a less severe clinical presentation and have been determined by a medical expert to require less laboratory testing. By contrast this tool is designed to identify patients at high risk for decompensation. 

As in the discussion above, median imputation is used during inference based on population statistics. This approach allows for patients with less than 6 lab values to be analyzed by the tool, with the acknowledgement that the prediction may be less accurate than patients with additional lab testing. 

6. What is the necessity and effect of normalizing the data?

Normalization of the data to similar scales facilitates numeric stability during the algorithm training process and ensures that all variables are initialized with relatively equal contribution to the prediction. This has been added to the methods. 

7. What were the criteria for including a variable in the model? 

Using a recursive feature selection technique, univariate statistical testing was applied to the cohort to identify the variables with greatest differences in distribution when stratified based on outcomes. Starting with the variable calculated to have the largest differences, additional variables are added to the model, one-by-one, in order of significance based on univariate testing, until the model performance plateaus. In this study, the optimal number of variables based on this technique was chosen to be 13. It should be noted that model performance did not degrade with additional variables (e.g. instead overall performance reached a plateau); thus the total number of variables used in this study represents the minimum amount needed to approximate the performance models using arbitrarily large number of covariates. This has been added to the methods.

8. I miss a Table 1 with all baseline characteristics (and difference between yes/no critical disease) and perhaps a table 2 with univariate logistic regression analyses of these parameters. 

Thank you for this comment. We have added Table 1 as well as the f-test data for table 2. 

The ranked f-tests are below: 

'Resp': 27.739717,

 'LDH': 23.125122,

 'Comorbidities': 16.79656,

 'BMI': 13.908264,

 'Ferritin': 12.375533,

 'WBC': 7.8667326,

 'Procalcitonin': 7.5904236,

 'Lymph': 7.47715,

 'Troponin': 7.263967,

 'CRP': 6.589011,

 'Creatinine': 5.7942467,

 'Age': 1.0775236,

 'Gender': 0.5919638

---

## [Decision Letter · Decision Letter 1]

13 Nov 2020

Development and External Validation of a Prognostic Tool for COVID-19 Critical Disease

PONE-D-20-14464R1

Dear Dr. Chow,

We’re pleased to inform you that your manuscript has been judged scientifically suitable for publication and will be formally accepted for publication once it meets all outstanding technical requirements.

Kind regards,

Itamar Ashkenazi

Academic Editor

PLOS ONE

Additional Editor Comments (optional):

The three reviewers differed in their opinion regarding this study. The main problem discussed with the reviewers was that this forecast model was constructed based on many variables and this model was validated on a rather small cohort. Another problem is that many of the parameters included in the forecast model rely on tests that are not in common use in the set up of acute care. I believe a note on these issues should be posted with the manuscript to be published. The authors could then reply. Whether more data has been accumulated (since this study was sent for publication) on this score in the authors' daily practice, presentation of these will be appreciated.

Reviewers' comments:

Reviewer's Responses to Questions

**Comments to the Author**

1. If the authors have adequately addressed your comments raised in a previous round of review and you feel that this manuscript is now acceptable for publication, you may indicate that here to bypass the “Comments to the Author” section, enter your conflict of interest statement in the “Confidential to Editor” section, and submit your "Accept" recommendation.

Reviewer #1: All comments have been addressed

Reviewer #2: All comments have been addressed

Reviewer #3: (No Response)

2. Is the manuscript technically sound, and do the data support the conclusions?

Reviewer #1: Yes

Reviewer #2: Partly

Reviewer #3: Yes

3. Has the statistical analysis been performed appropriately and rigorously? 

Reviewer #1: Yes

Reviewer #2: I Don't Know

Reviewer #3: No

4. Have the authors made all data underlying the findings in their manuscript fully available?

Reviewer #1: Yes

Reviewer #2: Yes

Reviewer #3: No

5. Is the manuscript presented in an intelligible fashion and written in standard English?

Reviewer #1: Yes

Reviewer #2: Yes

Reviewer #3: Yes

6. Review Comments to the Author

Reviewer #1: (No Response)

Reviewer #2: (No Response)

Reviewer #3: An important factor was ignored in the study—race. Disparity of COVID-19 between ethnic groups in US is profound.

7. PLOS authors have the option to publish the peer review history of their article (what does this mean?). If published, this will include your full peer review and any attached files.

Reviewer #1: No

Reviewer #2: **Yes: **Walter M. van den Bergh

Reviewer #3: No

---

## [Editor Report · Acceptance letter]

20 Nov 2020

PONE-D-20-14464R1 

Development and External Validation of a Prognostic Tool for COVID-19 Critical Disease 

Dear Dr. Chow:

I'm pleased to inform you that your manuscript has been deemed suitable for publication in PLOS ONE. Congratulations! Your manuscript is now with our production department. 

Kind regards, 

on behalf of

Dr. Itamar Ashkenazi 

Academic Editor

PLOS ONE